# Countering the advert effects of lung cancer on the anticancer potential of dendritic cell populations reinstates sensitivity to anti-PD-1 therapy

**Julyanne Brassard, Meredith Elizabeth Gill, Emilie Bernatchez, Véronique Desjardins, Joanny Roy, Philippe Joubert, David Marsolais, Marie-Renée Blanchet**[ID]*

Institut Universitaire de Cardiologie et de Pneumologie de Québec, Université Laval, Québec, Québec, Canada

* Marie-Renee.Blanchet@criucpq.ulaval.ca

**Data Availability Statement:** All relevant data are within the paper and its Supporting Information files.

## Abstract

Lung cancer is the leading cause of cancer-related deaths. While the recent use of immune checkpoint inhibitors significantly improves patient outcomes, responsiveness remains restricted to a small proportion of patients. Conventional dendritic cells (DCs) play a major role in anticancer immunity. In mice, two subpopulations of DCs are found in the lung: DC2s ($CD11b^+Sirp\alpha^+$) and DC1s ($CD103^+XCR1^+$), the latest specializing in the promotion of anti-cancer immune responses. However, the impact of lung cancer on DC populations and the consequent influence on the anticancer immune response remain poorly understood. To address this, DC populations were studied in murine models of Lewis Lung Carcinoma (LLC) and melanoma-induced lung metastasis (B16F10). We report that direct exposure to live or dead cancer cells impacts the capacity of DCs to differentiate into $CD103^+$ DC1s, leading to profound alterations in $CD103^+$ DC1 proportions in the lung. In addition, we observed the accumulation of $CD103^{lo}CD11b^+$ DCs, which express DC2 markers IRF4 and Sirpα, high levels of T-cell inhibitory molecules PD-L1/2 and the regulatory molecule CD200. Finally, DC1s were injected in combination with an immune checkpoint inhibitor (anti-PD-1) in the B16F10 model of resistance to the anti-PD-1 immune checkpoint therapy; the co-injection restored sensitivity to immunotherapy. Thus, we demonstrate that lung tumor development leads to the accumulation of $CD103^{lo}CD11b^+$ DCs with a regulatory potential combined with a reduced proportion of highly-specialized antitumor $CD103^+$ DC1s, which could promote cancer growth. Additionally, promoting an anticancer DC signature could be an interesting therapeutic avenue to increase the efficacy of existing immune checkpoint inhibitors.

## Introduction

Currently, lung cancer remains the most lethal cancer in industrialized countries. Despite significant advances in conventional therapies, the five-year survival rate remains lower than 20%

**Funding:** This work was supported by the Fonds sur les Maladies Respiratoires Bégin/Lavoie de l'Université Laval and by the Fondation de l'Institut Universitaire de Cardiologie et de Pneumologie de l'Université Laval to MRB (https://fondation-iucpq.org/). The funders had no role in study design, data collection and analysis, decision to publish, and preparation of the manuscript.

**Competing interests:** The authors have declared that no competing interests exist.

**Abbreviations:** DC, dendritic cell; FMO, fluorescence minus one; FLT3L, FMS-like tyrosine kinase 3 ligand; FLT3L-BMDC, FLT3L bone marrow-derived DCs; GM-CSF, granulocyte-macrophage colony-stimulating factor; ICI, immune checkpoint inhibitor; i.v., intravenously; LLC, Lewis lung carcinoma; LPS, lipopolysaccharide; MFI, mean fluorescence intensity; mo-DC, monocyte-derived DC; NSCLC, non-small cell lung carcinoma; PMA, Phorbol 12-myristate 13-acetate.

in most countries [1]. To support tumor development, cancer induces an immunoregulatory environment that reduces the anticancer function of immune cells [2]. Consequently, immunotherapies recently emerged as a new strategy to restore the natural antitumor immune response, and significantly improve survival. Immune checkpoint inhibitors (ICIs) that target the PD-1/PD-L1 axis are the most commonly used immunotherapy in patients with non-small cell lung carcinoma (NSCLC) and are now approved as first-line treatment in several countries [3, 4]. In normal conditions, the interaction between PD-L1 expressed on antigen-presenting cells and PD-1 present on T cells limits the T cell response to prevent auto-immunity. However, in cancer, PD-L1 is overexpressed by cancer cells and immune cells which are present within the tumor environment, leading to the inhibition of the cytotoxic T cells, which are crucial for the anticancer immune response [5]. While in some cases PD-1/PD-L1 inhibitors successfully restore the function of cytotoxic T cells and significantly improve patient survival, their effectiveness is limited to only a small proportion of patients [3, 6]. There is therefore an urgent need to better understand anticancer immune responses.

CD8 T cells or cytotoxic T cells are major effectors of ICIs and play an important role in the natural anticancer immune response. Indeed, CD8 T cell activation is initiated via antigen presentation by conventional dendritic cells (DCs) [7]. In the lung, DCs are a heterogeneous population, which in the past was divided according to surface marker expression, which can be highly variable based on the inflammatory context [8, 9] and differs between humans and mice [10]. Recently, DCs were thoroughly re-characterized based on cellular developmental pathways. This allowed the emergence of a new consensus in DC classification that better translates from mice to humans [10], where lung DCs comprise a mixture of CD103$^+$XCR1$^+$ (mice)/ CD141$^+$XCR1$^+$ (human) DC1s that depend on both BATF3 and IRF8 transcription factors for their development, and CD11b$^+$Sirpα$^+$ (mice and human) DC2s, which express the IRF4 transcription factor [10, 11]. DC1s are a major component of anticancer immune responses. Indeed, the absence of DC1 populations in *Batf3*$^{-/-}$ or *Irf8*$^{-/-}$ mice favours the growth of primary tumors or metastasis progression [12–14]. Furthermore, DC1s specialize in IL-12 production, trafficking of tumor antigens to draining lymph nodes and cross-presentation of tumor antigens to CD8 T cells [12–16]. Finally, DC1s also play an important role in immune checkpoints immunotherapies, as *Batf3*$^{-/-}$ mice do not respond to this type of treatment [17]. The contribution of DC2s in anticancer immune response is not well established, but some propose they are necessary to induce antitumor CD4 T cell immunity [18].

Despite this wealth of knowledge on anticancer immune responses, lung cancer immunotherapy remains weakly effective [3, 6]. This may stem in part from the current lack of knowledge on the impact of lung cancer on local DC populations, which are crucial in anticancer immunity. Previous studies by our group suggested that various inflammatory contexts profoundly impact the local DC signature, as well as disease progression [8, 9, 19]. Specifically, we demonstrated that the proportions of CD103$^+$ DC1s are drastically reduced under inflammatory conditions. We, therefore, set out to verify whether the development of lung cancer alters local DC populations, and whether enriching local DCs with high levels of anticancer CD103$^+$ DC1s could favourably impact the anticancer lung response. Using mouse models of lung cancer and melanoma-induced lung metastasis, we demonstrate that the cancer microenvironment decreases the proportions of anticancer CD103$^+$ DC1s. In return, we observed an unpredicted increase in a CD103$^{lo}$CD11b$^+$ DC population, which strongly expresses PD-L1/2 and CD200 regulatory molecules. Finally, we show that enriching the local DC population with CD103$^+$ DC1s supports a more efficient response to anti-PD-1s. These results suggest that lung tumor progression alters the local DC population signature to favour tumor growth and underline new mechanisms explaining the inability of the local DC1s to naturally regulate tumor growth, and possible resistance to anti-PD-1 therapies.

## Materials and methods

### Mice

C57Bl/6J mice were purchased from Jackson Laboratories and bred in a pathogen-free animal unit (*Centre de recherche de l'Institut Universitaire de Cardiologie et de Pneumologie de Québec*, *Université Laval*, *Québec*, QC, Canada). 8–12 weeks male and female mice were used for *in vivo* cancer models and 6–12 weeks male and female were used for *in vitro* protocols. Protocols were approved by local ethics committees and followed Canadian animal care guidelines.

### *In vivo* tumor models

For the induction of lung tumor models, mice were injected intravenously (i.v.) with 2.5 x $10^5$ B16F10 melanoma cells (ATCC, catalog no. CRL-6475) or $10^6$ Lewis lung carcinoma (LLC) cells (ATCC, catalog no. CRL-1642), previously grown in DMEM media (Wisent) supplemented with 10% FBS (Wisent). 18 days following cancer cells injection, mice were euthanized and lungs were collected. An arbitrary cancer score indicative of the number and size of tumors was fixed from 0 (no visible tumor) to 5 (highest score). In the B16F10 lung metastasis model, 200 μg/mice anti-mouse PD-1 antibody (BioXcell, catalog no. BP0033-2) or 200 μg/mice Armenian hamster IgG isotype control (BioXcell, catalog no. BP0091) were administered via intraperitoneal injection on day 4, 8, 11 and 14 in combination with an i.v. injection of 3 x $10^5$ XCR1[+] FLT3L-BMDCs on day 0, 4, 8, 11 and 14 (Fig 7C) [20–22].

### Production of FLT3L bone marrow-derived DCs (FLT3L-BMDCs)

Bone marrow cells were isolated by flushing marrow from tibias and femurs using a 27-gauge needle and PBS. Cells were cultured at 1.5 x $10^6$ cells/ml for 7 days in RPMI 1640 media (Wisent) supplemented with 10% FBS (Wisent), 50 μM β-mercaptoethanol, antibiotic/antimycotic (Wisent) and 100 ng/ml FMS-like tyrosine kinase 3 ligand (FLT3L) (Peprotech, catalog no. 250-31L). On day 7, 10 ng/ml of Granulocyte-macrophage colony-stimulating factor (GM-CSF) (Peprotech, catalog no. 315–03) was added to the culture. For the stimulation of FLT3L-BMDCs, $10^4$ live or an antigenic preparation (obtained by two cycles of freeze and thaw) of B16F10 or LLC cells per million of DCs were added. In some stimulations, DCs were segregated from live cancer cells with a 0.4 μm cell culture insert (Falcon). For transfer experiments, FLT3L-BMDCs were stimulated on day 7 with GM-CSF and on day 9 with live B16F10 cells and harvested on day 10. Before the injection, XCR1[+] FLT3L-BMDCs were isolated using XCR1-APC (Biolegend) and EasySep™ Mouse APC Positive Selection Kit II (Stemcell).

### FLT3L-BMDCs phagocytosis assay

B16F10 cells were stained for 20 minutes using the CellTrace™ CFSE Cell Proliferation Kit (Invitrogen) according to the manufacturer's instructions, and washed. $10^4$ CFSE-B16F10 cells per million of DCs were then used to stimulate FLT3L-BMDCs for 24h. DC CFSE expression was measured by flow cytometry.

### Flow cytometry

For flow cytometry analysis, the lung tissue was digested with 200 U/mL Collagenase IV (Sigma-Aldrich) for 45 min at 37˚C and pressed through a 70 μm cell strainer. Red blood cells were lysed with ammonium chloride solution. Antibodies used were CD103-PE, CD103-biotin, CD103-APC-Cy7, CD11c-BV711, CD11c-BV785, I-A/I-E (MHC II)-Pacific Blue, CD172a (Sirpα)-APC-Cy7, CD19-biotin, CD90.2-biotin, CCR2-biotin, Ly-6C-APC-Cy7, IRF4-PE, CD86-APC-Cy7, CD197 (CCR7)-APC, TGF-β1-APC, TNF-APC-Cy7, H-2K[b]/H-2D[b] (MHC

I)-APC, CD200-PE, PD-L1-APC, PD-L2-PE, CD366 (TIM-3)-APC, IL-12/IL-23 p40-APC, XCR1-APC, XCR1-BV650 (Biolegend), NK1.1-biotin (Ablab), CD11b-Pe-Cy7, CD103-PE, Sirpα-BV711, Zbtb46-PE (BD Bioscience), CD11b-AF700 (eBioscience), IRF8-APC (Miltenyi Biotec) and CD80-biotin (BD PHARMINGEN). For cytokine intracellular staining, lung-isolated cells or FLT3L-BMDCs were stimulated for 4h with 50 ng/ml Phorbol 12-myristate 13-acetate (PMA) (Sigma-Aldrich), 500 ng/ml Ionomycin (Sigma-Aldrich) and 10 μg/ml brefeldin A (Sigma-Aldrich) at 37°C. Intracellular staining was performed using the True-Nuclear™ Transcription Factor Buffer Set (Biolegend) according to the manufacturer's instructions. Cells were analyzed using a BD LSR Fortessa cytometer (BD Biosciences) and FlowJo software V10 (BD). Doublets were discarded from the analysis by sequentially selecting the linear population from FSC-A/FSC-H and SSC-A/SSC-H dot plots. When indicated, autofluorescent cells were removed from the analysis using the FITC channel. At least, $2 \times 10^5$ lung-isolated cells and $3 \times 10^4$ FLT3L-BMDCs were processed. Mean fluorescence intensity (MFI) data were analyzed as Δ MFI, which corresponds to the MFI of the antigen-positive population minus the MFI of the fluorescence minus one (FMO) control of this population.

## Statistics

Data are presented as mean ± SEM. Data were tested for normality and homogeneity of variance using GraphPad Prism software, and the required statistical analysis was performed according to the normality of data, as suggested by the software. Accordingly, statistical analysis for multiple comparisons was performed using an ANOVA table followed by Tukey's multiple comparison tests. Non-multiple comparisons were analyzed using paired or unpaired t-tests. Statistical significance was determined at $p < 0.05$.

## Results

### Cancer development decreases the proportions of lung CD103+XCR1+ DC1s

To evaluate the impact of the lung cancer environment on DC populations, two different cancer models were used. The first (LLC) is an orthotopic model of lung cancer that develops as a squamous cell carcinoma. The second one uses B16F10 melanoma cells and is a pulmonary metastatic model. The development of lung tumors resulted in an increased lung index (lung weight/mice weight) and total lung cell numbers in both cancer models (Fig 1A) compared to naïve mice. Total DCs were characterized (as previously published by our group [8, 9]) as auto-fluorescent-, CD90.2-, CD19-, NK1.1-, MHC II+ and CD11c+ (see Fig 1B for DC gating). The presence of LLC and B16F10-induced tumors strongly impacted the relative proportions of these lung DC populations. The percentage of lung CD103+XCR1+ DC1s gradually decreased and reciprocally, the percentage of CD11b+Sirpα+ DC2s increased slightly in both models (Fig 1C–1E). These results indicate that tumor development impacts the balance between CD103+XCR1+ DC1s and CD11b+Sirpα+ DC2s at the expense of the antitumor DC1 population.

### Cancer cells/antigens directly inhibit CD103+XCR1+ DC1 differentiation

We previously demonstrated that antigens or inflammatory molecules such as lipopolysaccharide (LPS) and TNF interfere with CD103 expression on DC1 and alter DC1 differentiation [8]. We thus tested whether cancer cells (alive or dead) directly impact the capacity of DC precursors to differentiate into CD103+ DC1s, which could in part explain the altered DC populations observed in Fig 1. FLT3L-BMDCs were stimulated with GM-CSF to induce DC1 CD103 expression [8, 23], and exposed to LLC or B16F10 cells (live or an antigenic preparation of

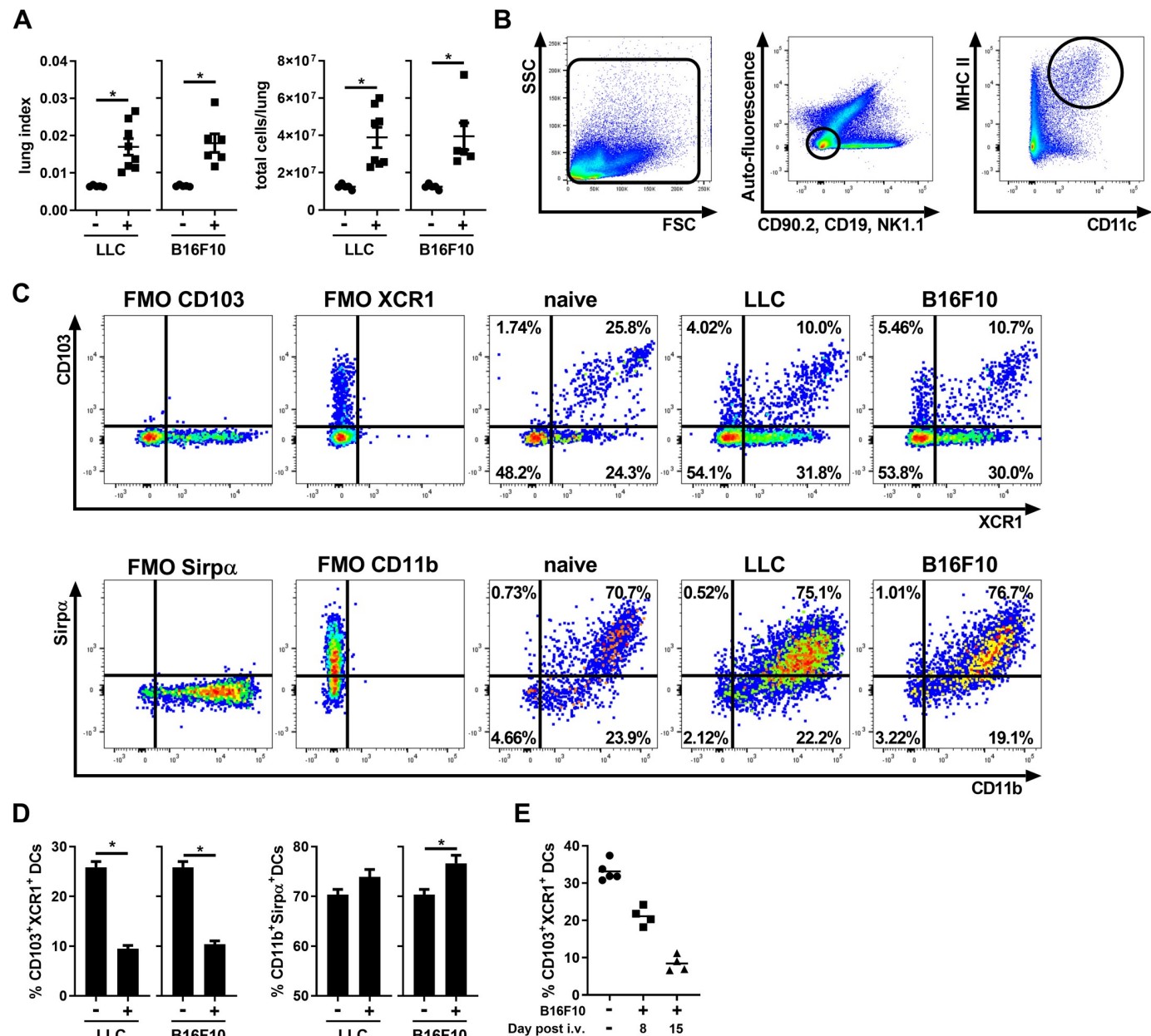

**Fig 1. Lung tumor development decreases the proportions of CD103$^+$XCR1$^+$ DC1s.** Analysis of lung tumors and DC populations following i.v. injection of B16F10 or LLC cancer cells. (A) Lung index (lung weight/mice weight) and total lung cell number. (B) Gating strategy for the identification of total DCs. DCs were gated on auto-fluorescence$^-$, NK1.1$^-$, CD90.2$^-$, CD19$^-$, MHC II$^{Hi}$ and CD11c$^+$. (C) Representative flow cytometry profile of DC expression of CD103 x XCR1 and Sirpα x CD11b analysis based on fluorescence minus one (FMO) controls. (D) Percentage of CD103$^+$XCR1$^+$ DC1 and CD11b$^+$Sirpα$^+$ DC2 of MHC II$^{hi}$CD11c$^+$ cells (DCs). (E) Percentage of CD103$^+$XCR1$^+$ DC1 at 8 and 15 days after the i.v. injection of B16F10 cells. (A-D) Data are expressed as mean ± SEM. n = 5–8 mice per group and are representative of 2–6 independent experiments. $^*$ = p < 0.05 using (A-D) an unpaired t-test. (E) Data are presented as individual points with means.

cancer cells). As expected, GM-CSF alone increased the percentage of CD103$^+$XCR1$^+$ DCs in cultures (Fig 2A and 2B). However, the addition of live LLC or B16F10 cells, or exposure to an antigenic preparation during GM-CSF stimulation significantly decreased the percentage of CD103$^+$XCR1$^+$ DC compared to GM-CSF alone (Fig 2A and 2B). Additionally, this phenomenon was significantly reversed when DCs were segregated from LLC and B16F10 using 0.4 μm inserts (Fig 2C), indicating that the contact between DCs and cancer cells is necessary to

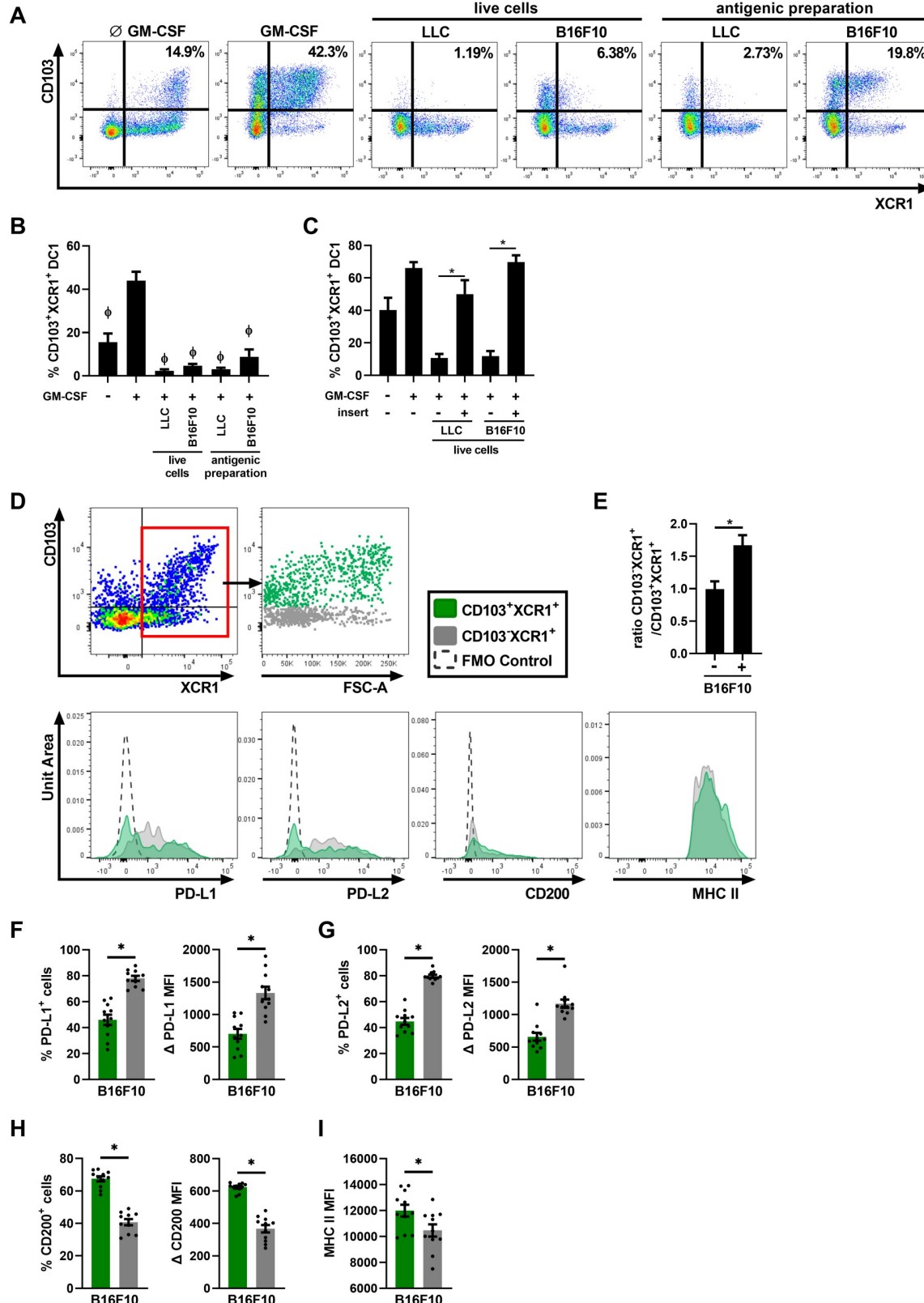

**Fig 2. Cancer cells prevent the differentiation of bone marrow precursors into CD103+XCR1+ DC1s.** (A-C) FLT3L-BMDCs were exposed to GM-CSF ± live B16F10/LLC cells or a B16F10/LLC antigenic preparation (A and B). In C), DCs were segregated (or not)

from cancer cells with a 0.4 µm insert. (A) Representative flow cytometry profiles of CD103 and XCR1 expression on DCs. (B-C) Percentage of CD103$^+$XCR1$^+$ of MHC II$^{Hi}$CD11c$^+$ DCs. Data are expressed as mean ± SEM. n = 5, pooled from two independent experiments. φ = p < 0.05 compared to GM-CSF alone condition, $^*$ = p < 0.05 when conditions with and without inserts are compared. P-values were determined using repeated measures one-way ANOVA, with the Geisser-Greenhouse correction followed by Tukey's multiple comparisons test. (D to I) Lung DC populations were analyzed by flow cytometry following an i.v. injection of B16F10 cancer cells. (D) Gating strategy for the identification of CD103$^+$XCR1$^+$ and CD103$^-$XCR1$^+$ populations from previously gated autofluorescent$^-$, CD19$^-$, CD90.2$^-$, CD20$^-$, MHC II$^+$, CD11c$^+$ DCs and representative histogram of PD-L1, PD-L2, CD200, MHC II within these two populations. (E) The ratio of the number of CD103$^-$XCR1$^+$ DCs over CD103$^+$XCR1$^+$ DCs. Percentage and Δ MFI of (F) PD-L1, (G) PD-L2, (H) CD200 and (I) MHC II of CD103$^+$XCR1$^+$ DCs and CD103$^-$XCR1$^+$ in mice injected with B16F10 cells. Data are expressed as mean ± SEM. n = 11 pooled from two independent experiments. $^*$ = p < 0.05 using two-way ANOVA with Šídák's multiple comparisons test.

prevent CD103$^+$ DC1 differentiation. These results suggest that cancer cells, or an antigenic mix of dead cancer cells could directly alter the proportions of CD103$^+$ DC1s *in vivo*.

In the lung, CD103 is one of the main markers used to identify DC1s. Recently, other markers such as XCR1 have also been used to identify this population [10, 11]. Using XCR1 to stain the DC1 population, we observed that following B16F10 lung metastasis development, the ratio of lung CD103$^-$XCR1$^+$ over CD103$^+$XCR1$^+$ DCs was significantly increased compared to naïve mice, suggesting an accumulation of DC1s that do not express CD103 (Fig 2E). As CD103 is normally used to identify DC1s, the difference of function between CD103$^+$XCR1$^+$ DCs and CD103$^-$XCR1$^+$ DCs in cancer is not well-established. We therefore analyzed the expression of PD-L1 and PD-L2, two regulatory molecules that induce the inhibition of T cell proliferation, survival and effector functions through their binding with PD-1 on T cells [5]. While both populations express PD-L1 and PD-L2, the percentage of positive cells and the MFI for these two inhibitory molecules were significantly higher within the CD103$^-$ population following tumor development (Fig 2D, 2F and 2G). Conversely, the expression (percentage and MFI) of CD200, another regulatory molecule, was significantly higher on CD103$^+$ cells compared to CD103$^-$XCR1$^+$ DCs (Fig 2D and 2H) [24]. Finally, the MFI of MHC II, which is a marker of DC maturation and activation, was found at a high level in both populations, with MFIs over 10 000 units, but slightly higher in CD103$^+$XCR1$^+$ compared to CD103$^-$ DCs (Fig 2D and 2I) [25]. This demonstrates that following cancer development, the lung DC signature is skewed towards CD103$^-$ DC1s with high regulatory and activation potential.

## Tumor development leads to the accumulation of CD103$^{lo}$CD11b$^+$ DCs in the lung

Through the thorough dissection of the local DC population signature in cancer, we observed that the CD103$^+$ DC1 (population circled in green, Fig 3A) and CD11b$^+$ DC2 (population circled in blue, Fig 3A) are well segregated in naïve mice. However, following the injection of either LLC or B16F10 cells, a third DC population (circled in red) co-expressing low to intermediate levels of CD103 and CD11b was also observed (Fig 3A). We termed this population CD103$^{lo}$CD11b$^+$ DCs. This population was significantly increased in the lung for both cancer models (Fig 3B). The ratio of anticancer DC1s to CD103$^{lo}$CD11b$^+$ DCs was reduced from approximately 4: 1 in naïve mice to 1: 1 in both cancer models (Fig 3C), indicating that the number of CD103$^{lo}$CD11b$^+$ DCs is similar to that of anticancer CD103$^+$ DC1s following tumor development in the lung.

## CD103$^{lo}$CD11b$^+$ DCs express surface markers and transcription factors that are characteristic of a DC2 population

CD103$^+$CD11b$^+$ DC2 populations were reported in the gut in various models [26, 27], but whether the CD103$^{lo}$CD11b$^+$ DCs we observed following cancer development were functionally similar to gut CD103$^+$ DC2s remained unclear. Additionally, since lung MHC II$^+$CD11c$^+$

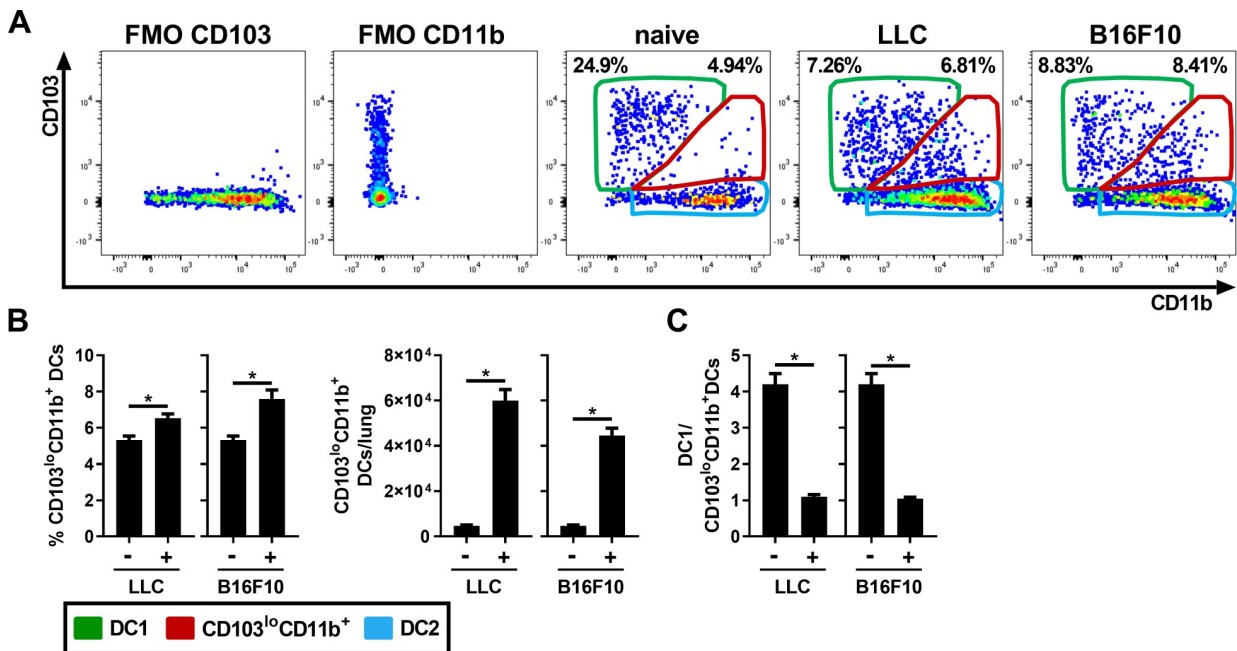

**Fig 3. Accumulation of CD103^loCD11b+ DCs following lung cancer development.** Lung DC populations were analyzed by flow cytometry following an i.v. injection of LLC and B16F10 cancer cells. (A) Representative flow cytometry profiles of CD103 and CD11b expression on MHC II^hiCD11c+ DCs, showing conventional DC1 population in green, DC2 population in blue and CD103^loCD11b+ DCs in red. (B) Percentage and number of CD103^loCD11b+ DC. (C) Ratio of CD103+CD11b^-/lo DC1 (green) on CD103^loCD11b+ (red) DCs. Data are expressed as mean ± SEM. n = 5–8 mice per group and are representative of two independent experiments. * = p < 0.05 using an unpaired t-test with Welch's correction.

DCs expressing CD11b can originate from bone marrow pre-DCs (conventional DCs), but also blood monocytes (monocyte-derived DCs (mo-DCs)), the origin of this CD103^loCD11b+ DC population was ambiguous [11, 28]. Thus, two surface markers, CCR2 and Ly-6C, which are respectively associated with the monocyte lineage and mo-DCs, were analyzed on total MHC II+CD11c+ DCs, and compared between the CD103+ DC1, CD103^loCD11b+ DC and total CD11b+ DC2 (which include mo-DCs) populations (the gating strategy specific to this section is presented in S1 Fig) [28, 29]. The percentage of CD103^loCD11b+ DCs expressing CCR2 was similar to CD11b+ DC2s following LLC injection (Fig 4A). In contrast, in response to B16F10 injection, the percentage of CD103^loCD11b+ DCs expressing CCR2 was similar to DC1s, i.e. fairly low (Fig 4A). However, in both cancer models, the percentage of CD103^loCD11b+ DCs expressing Ly-6C, a robust marker of mo-DCs, was significantly lower than CD11b+ DC2s (Fig 4B). This suggests that a significant proportion of CD103^loCD11b+ DCs is derived from pre-DCs and does not originate from the monocyte lineage. To confirm the pre-DC origin, we analyzed the expression of ZBTB46, a transcription factor expressed by pre-DCs and conventional DC populations, which was found in the vast majority of CD103^loCD11b+ DCs (Fig 4C) [30].

To further assess whether CD103^loCD11b+ DCs are associated with the DC1 or DC2 conventional DC subsets, XCR1 and Sirpα surface expression was analyzed and compared to DC1 or DC2 conventional DC populations. We observed that the vast majority of the CD103^loCD11b+ DC population (Fig 4D; red population), co-distributes with CD11b+ DC2s on the XCR1 vs Sirpα contour plots. Also, CD103^loCD11b+ DCs express Sirpα at a similar level to DC2s, and XCR1 at a significantly lower level than DC1s. To deepen the characterization of CD103^loCD11b+ DCs, IRF8 (DC1) and IRF4 (DC2) transcription factors expression were also compared between DC populations. As observed in the IRF4 and IRF8 contour plots

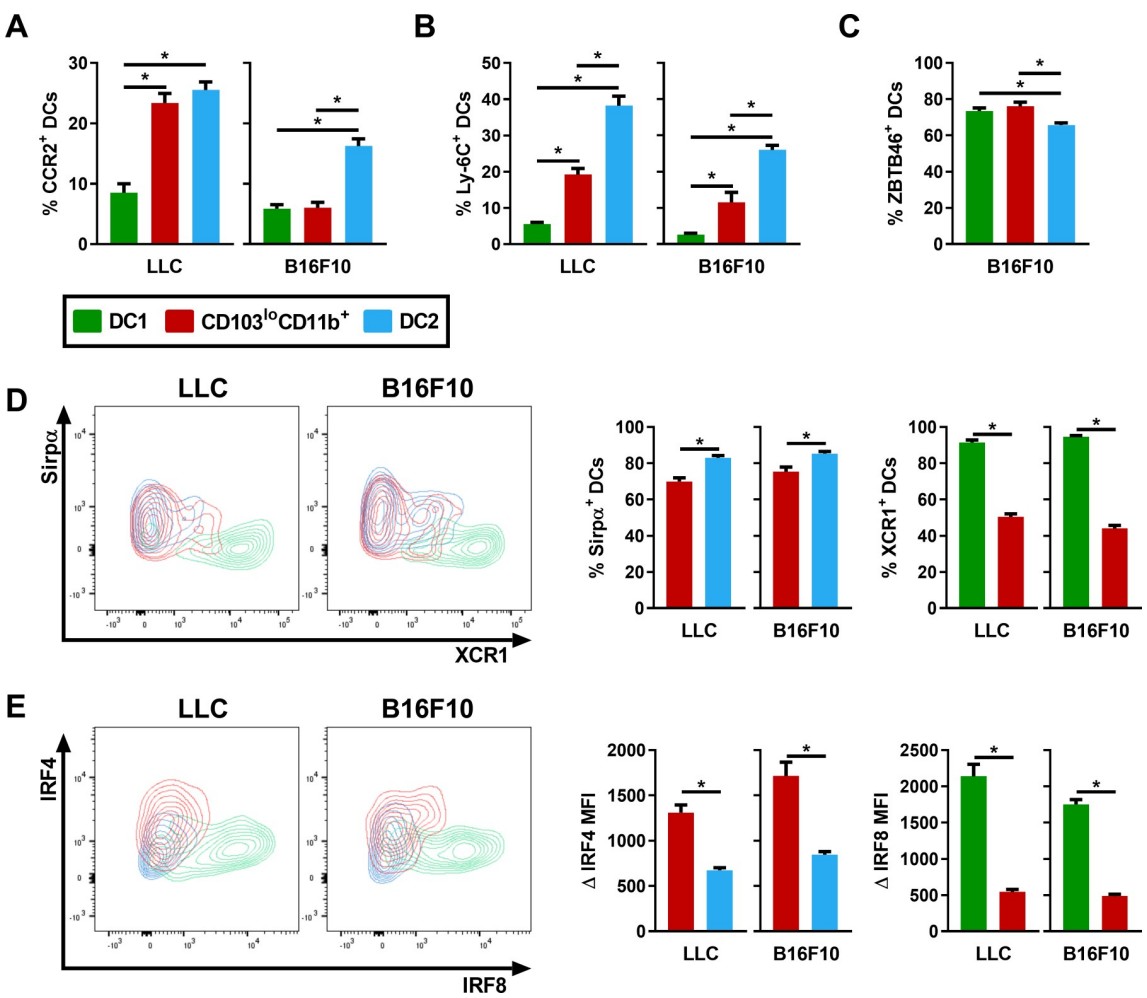

**Fig 4. CD103$^{lo}$CD11b$^+$ DCs express markers of the DC2 population.** Surface markers and transcription factors expression were analyzed by flow cytometry and compared between lung DC1 (green), CD103$^{lo}$CD11b$^+$ DC (red) and DC2 (blue) populations following the i.v. injection of LLC or B16F10 cells. Percentage of (A) CCR2$^+$, (B) Ly-6C$^+$ and (C) ZBTB46$^+$ cells for each subpopulation of DCs. (D) Representative contour plots of Sirpα and XCR1 expression of each DC subpopulation, as well as the percentage of Sirpα$^+$ cells in CD103$^{lo}$CD11b$^+$ DCs (red) and DC2s (blue), and the percentage of XCR1$^+$ cells in DC1s (green) and CD103$^{lo}$CD11b$^+$ DCs (red). (E) Representative contour plots of IRF4 and IRF8 expression, as well as Δ IRF4 MFI in CD103$^{lo}$CD11b$^+$ DCs (red) and DC2s (blue), and Δ IRF8 MFI in DC1s (green) and CD103$^{lo}$CD11b$^+$ DCs (red). Data are expressed as mean ± SEM. (A-B-E) n = 5–8 mice per group and are representative of two independent experiments. (C-D) n = 11–14 pooled from two independent experiments. * = p < 0.05, (A-B-C) using repeated measures one-way ANOVA, with the Geisser-Greenhouse correction followed by Tukey's multiple comparisons test and (D-E) using a paired t-test.

(Fig 4E), CD103$^{lo}$CD11b$^+$ DCs co-distributes with the DC2 population. The MFI of IRF4 and IRF8 was compared between CD103$^{lo}$CD11b$^+$ DCs and DC1s/DC2s. The IRF4 MFI in CD103$^{lo}$CD11b$^+$ DCs was significantly higher than conventional DC2s. Additionally, IRF8 expression was significantly lower than DC1s in this population (Fig 4E). Therefore, the surface markers and transcription factors analyzes indicate that CD103$^{lo}$CD11b$^+$ DCs are likely associated with the DC2 population in these models.

## CD103$^{lo}$CD11b$^+$ DCs express high levels of migratory, co-stimulatory and antigen-presenting molecules

With a presence in the lung that is quantitatively comparable to that of anticancer CD103$^+$ DC1s, we set out to address the capacity of CD103$^{lo}$CD11b$^+$ DCs to present antigen and

migrate, as indicators of their functional potential. To do so, we verified the expression of antigen presentation molecules, chemokine receptors involved in DC migration as well as co-stimulatory molecules.

MHC I participates in the cross-presentation of tumor antigens by DCs to CD8 T cells, while MHC II is involved in antigen presentation to CD4 T cells [7]. We observed that CD103$^{lo}$CD11b$^{+}$ DCs strongly express MHC I and MHC II in both models, to a level similar or higher than other DC subpopulations (Fig 5A). CCR7 is involved in DC trafficking from the lung to the draining lymph nodes [7]. We observed that the CD103$^{lo}$CD11b$^{+}$ population expresses higher CCR7 levels than the other DC populations in both cancer models (Fig 5B), suggesting a strong potential for trafficking to the lymph nodes and T cell interactions. Additionally, in LLC and B16F10 models, the co-stimulatory molecule CD80 surface expression was higher in CD103$^{lo}$CD11b$^{+}$ DCs compared to the CD103$^{+}$ DC1 population, (Fig 5C). Alternatively, the co-stimulatory molecule CD86 expression was significantly higher in CD103$^{lo}$CD11b$^{+}$ DCs than in the DC2 population (Fig 5C).

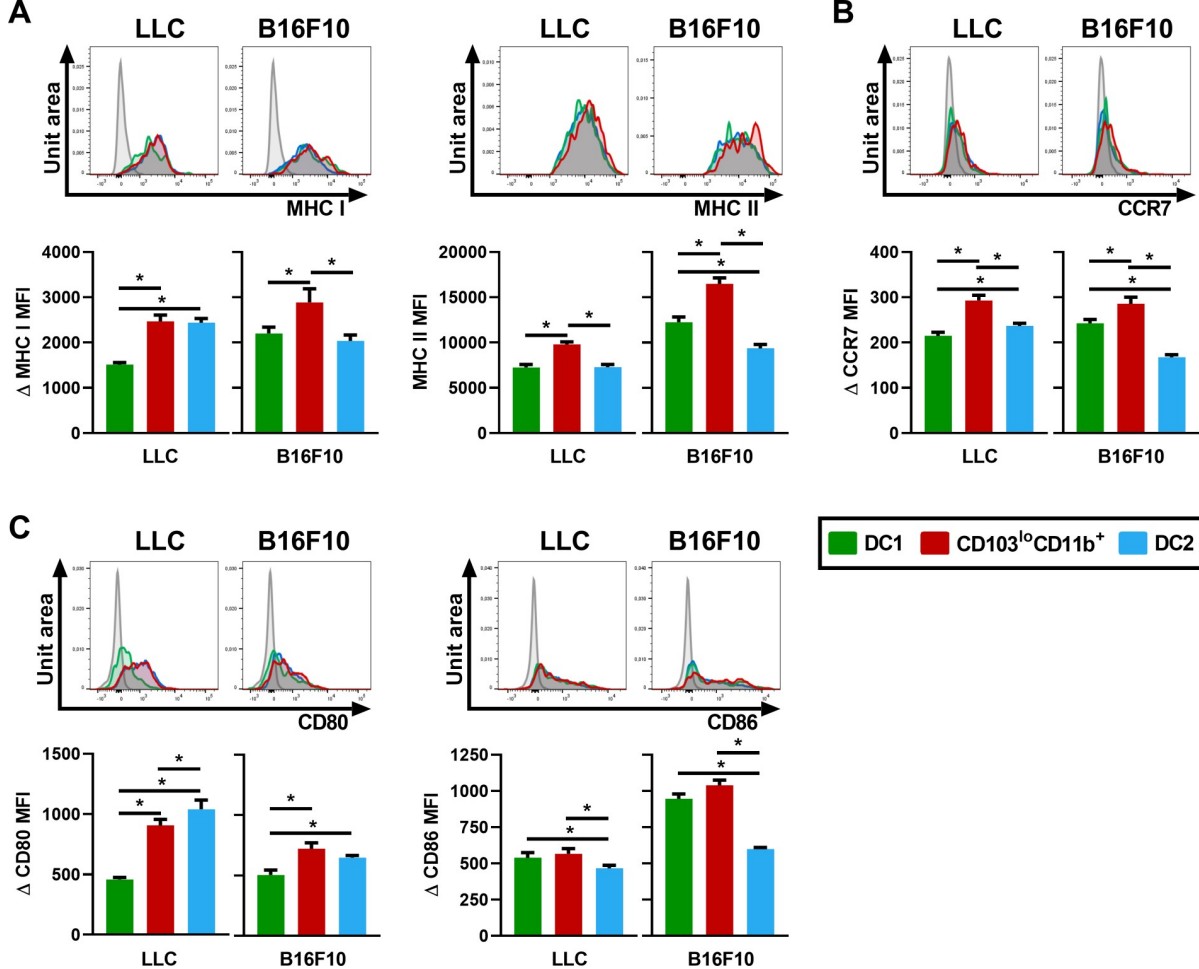

**Fig 5. CD103$^{lo}$CD11b$^{+}$ DCs are activated and show strong potential for T cell interactions.** Expression of surface markers and cytokines production were analyzed by flow cytometry and compared between lung DC subpopulations following the i.v. injection of LLC or B16F10. Upper panels in (A-B-C): representative flow cytometry histograms showing the normalized number of cells (unit area) on the Y-axis and fluorescence intensity on the X-axis. FMO controls appear in grey in each histogram. Lower panels in (A-B-C): comparison of (A) ΔMHC I, MHC II, (B) ΔCCR7, (C) ΔCD80 and ΔCD86 MFI between DC1 (green), CD103$^{lo}$CD11b$^{+}$ DC (red) and DC2 (blue) populations. Data are expressed as mean ± SEM. n = 10–17 pooled from two independent experiments. * = p < 0.05 using repeated measures one-way ANOVA, with the Geisser-Greenhouse correction followed by Tukey's multiple comparisons test.

All and all, this cluster of results suggests that in cancer, CD103$^{lo}$CD11b$^{+}$ DCs are overall profiled to present antigen, co-stimulate T cells upon antigen presentation and migrate to lymph nodes compared to CD103$^{+}$ DC1s and DC2s.

## The CD103$^{lo}$CD11b$^{+}$ DC population expresses high levels of regulatory molecules and produces low levels of IL-12

The regulatory processes allowing tumor progression are linked to the induction/ production of regulatory molecules by DCs, such as TGF-β, PD-L1, PD-L2 and CD200 [5, 24]. Indeed, the interaction of PD-L1/2 with PD-1 on T cells negatively impacts immune responses through the inhibition of T cell proliferation, survival and effector functions [5]. DCs can also produce the regulatory cytokine TGF-β while CD200 interacts with its receptor to induce an inhibitory signal preventing activation [31]. Finally, TIM-3 was recently shown to exert regulatory functions when expressed by CD103$^{+}$ DC1 [32]. These regulatory markers were therefore evaluated in CD103$^{lo}$CD11b$^{+}$ DCs to determine the regulatory potential of this population.

Following lung tumor development, all three DC populations expressed PD-L1, but CD103$^{lo}$CD11b$^{+}$ DCs expressed the highest levels of PD-L1 (Fig 6A). We also noted that an important proportion of CD103$^{lo}$CD11b$^{+}$ DCs expressed PD-L2 (whereas few CD103$^{+}$ DC1s and DC2s did), with a significantly higher MFI than DC1 and DC2s (Fig 6B). Therefore, both PD-L1 and PD-L2 are highly expressed on CD103$^{lo}$CD11b$^{+}$ DCs in lung cancer. While we observed a higher CD200 expression on CD103$^{lo}$CD11b$^{+}$ DCs compared to DC1s and DC2s in both models (Fig 6C), TGF-β expression was significantly higher in CD103$^{lo}$CD11b$^{+}$ DCs compared to DC1s following LLC injection only (Fig 6D). Finally, TIM-3 expression was significantly higher in the CD103$^{+}$ DC1 population compared to CD103$^{lo}$CD11b$^{+}$ DCs and DC2s (Fig 6E). Independently of the TIM-3 expression, the significantly higher expression of PD-L1, PD-L2 and CD200 combined with a higher TGF-β production by CD103$^{lo}$CD11b$^{+}$ DCs compared to other DC populations bestow a convincing immunoregulatory potential to this population.

Previous studies demonstrated that CD103$^{+}$ DC1s are the main producer of IL-12, an important step in cancer management by immune cells [12, 13]. We observed that IL-12 p40 production in CD103$^{lo}$CD11b$^{+}$ DCs was significantly lower than CD103$^{+}$ DC1s, but slightly higher than DC2s (Fig 6F). These results, combined with the high expression of regulatory molecules, further supports the idea that CD103$^{lo}$CD11b$^{+}$ DCs do not exert an effective anti-cancer immune response and rather likely contribute to the immunoregulatory environment in cancer.

## Enrichment with CD103$^{+}$ DC1s improves anti-PD-1 sensitivity

Most current immunotherapy strategies focus on improving T cell function by targeting ICI pathways [3]. However, we demonstrate that lung tumors influence the local DC population signature by decreasing the anticancer CD103$^{+}$ DC1s proportions and inducing an accumulation of CD103$^{lo}$CD11b$^{+}$ DCs with regulatory potential. These two phenomena likely cooperate to blunt anticancer immunity. However, we also propose that local DC populations in cancer are ill-equipped to present cancer antigen, which could explain the mitigated success of ICIs, which relies on the assumption of efficient interactions between DCs and T cells.

Therefore, we wondered whether enriching the local DC population with CD103$^{+}$ DC1s, specialized in presenting cancer antigen, would enhance the response to ICIs. To address this question, we first set out to generate a large amount of anticancer FLT3L-BMDCs, and prime these cells with tumor antigen. Our first technical challenge was to counteract the downregulating impact of cancer cell exposure on CD103$^{+}$ DC1 differentiation. We therefore determined that adding cancer cells two days following GM-CSF allowed for maximal

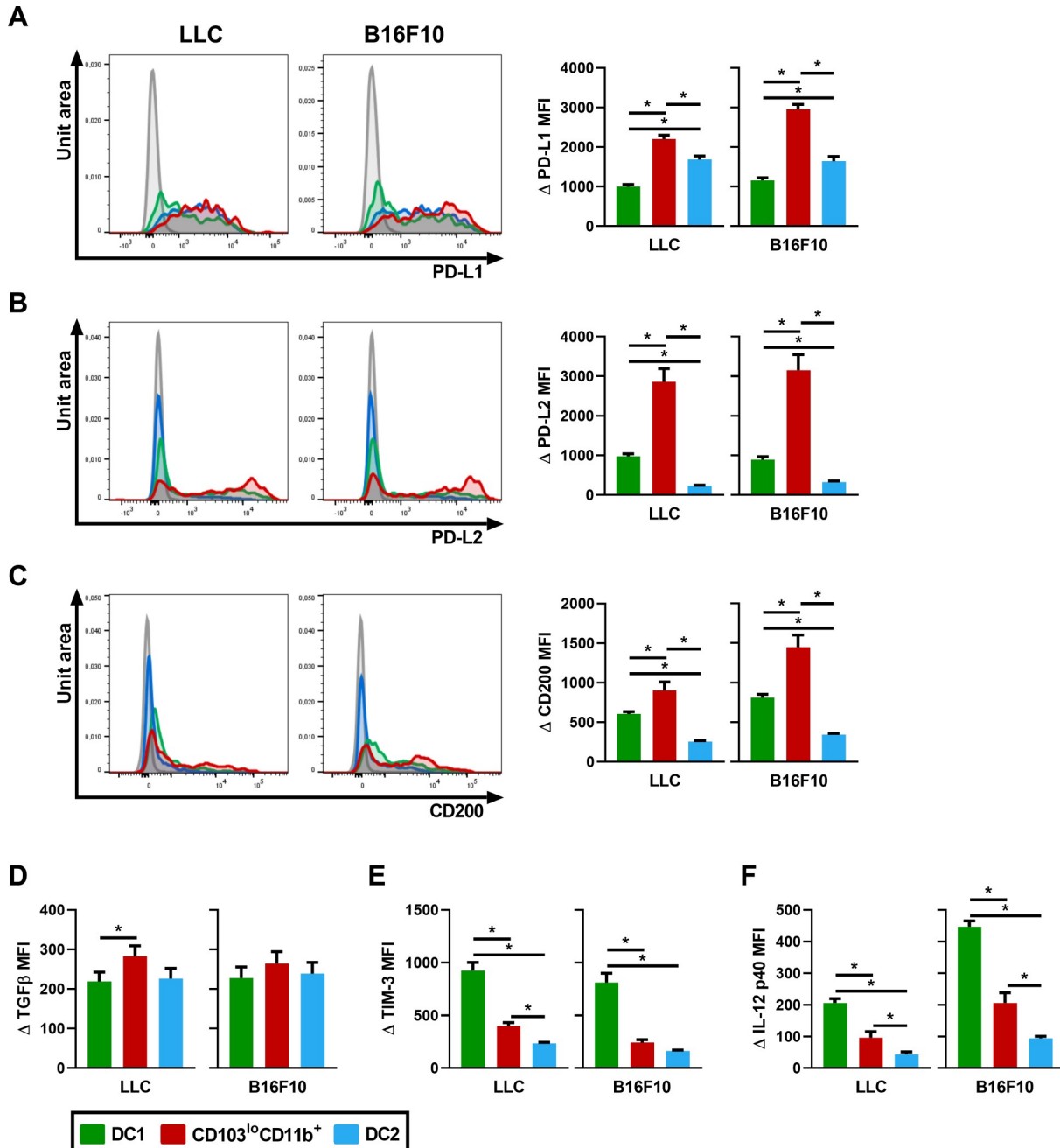

**Fig 6. CD103^loCD11b^+ DCs express regulatory molecules.** Surface marker expression and cytokines production were analyzed by flow cytometry and compared between lung DC subpopulations following the i.v. injection of LLC or B16F10. Left panels on (A-B-C): representative flow cytometry histograms showing the normalized numbers of cells (unit area) on the Y-axis and fluorescence intensity on the X-axis for (A) PD-L1, (B) PD-L2 and (C) CD200. FMO control appears in grey in each histogram. Right panels on (A-B-C) and (D-E-F): comparison of (A) ΔPD-L1, (B) ΔPD-L2, (C) ΔCD200, (D) ΔTGFβ, (E) ΔTIM-3 and (F) ΔIL-12 p40 MFI MFI between DC1 (green), CD103^loCD11b^+ DC (red) and DC2 (blue) populations. Data are expressed as mean ± SEM. n = 10–17 pooled from two independent experiments. * = p < 0.05 using repeated measures one-way ANOVA, with the Geisser-Greenhouse correction followed by Tukey's multiple comparisons test.

differentiation of CD103^+XCR1^+ DC1s in the presence of cancer cells (S2A Fig). Critically, the anticancer cytokine IL-12 was strongly induced when DCs were stimulated with live cancer cells (not with the antigenic preparation) two days after GM-CSF stimulation (Fig 7A). This

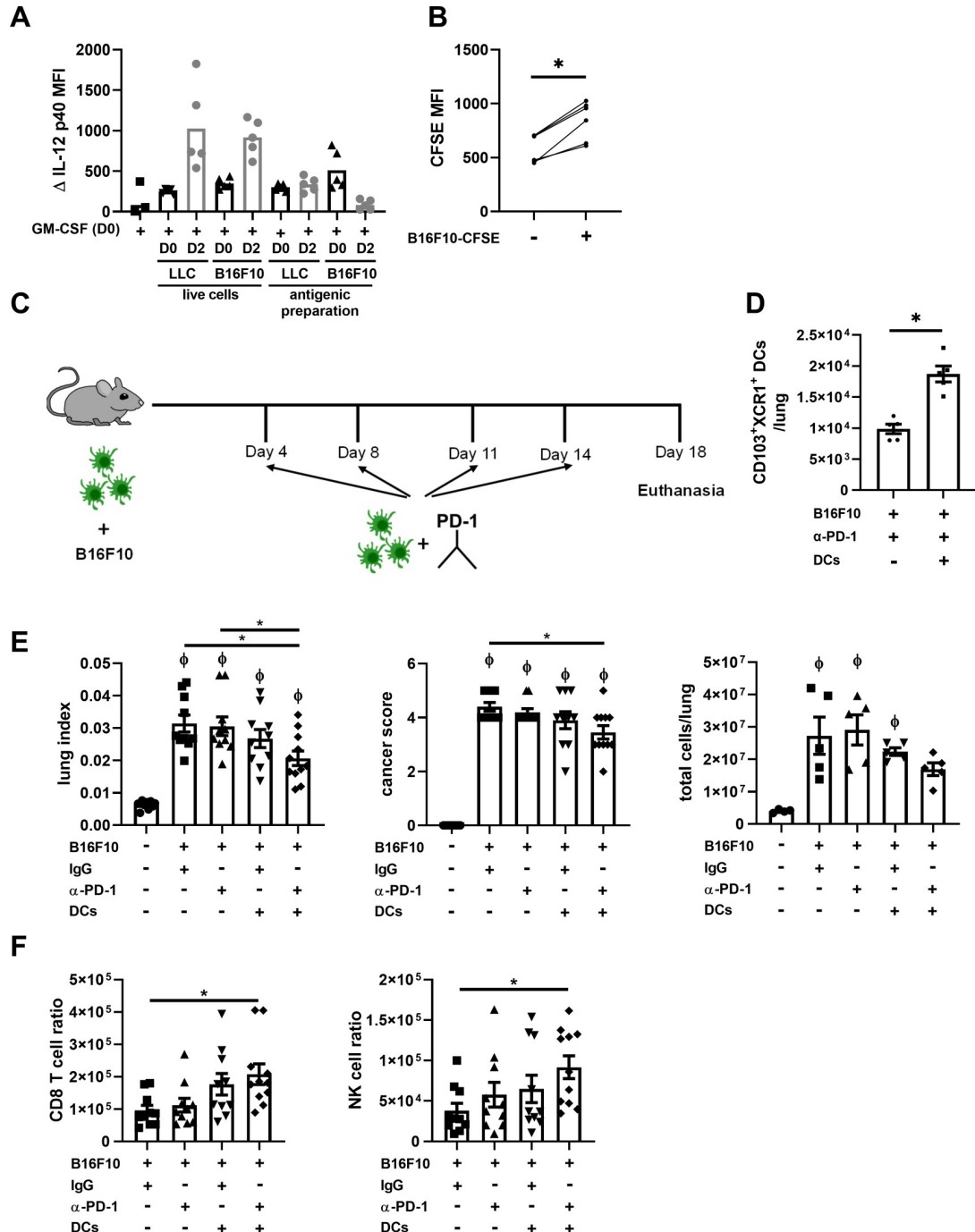

**Fig 7. Injection of XCR1⁺ DC1 improves sensitivity to anti-PD-1 treatment.** (A) FLT3L-BMDCs were stimulated with GM-CSF (Day 0). Either live B16F10/ LLC cells or a B16F10/ LLC antigenic preparation was added on day 0 or day 2 and ΔIL-12 p40 MFI was measured by flow cytometry on day 3. (B) Two days following GM-CSF stimulation FLT3L-BMDCs were stimulated for 24h with CFSE-treated B16F10 cells and CFSE MFI was measured by flow cytometry in CD11c⁺MHC II⁺ DC ± stimulation with CFSE-B16F10. (C) Schematic representation of the timeline treatment with the anti-PD-1 and XCR1⁺ DC1 injections (D-E-F) Analysis of mice lungs (D) 9 days or (E-F) 18 days after B16F10 injections. (D) The total number of lung CD103⁺XCR1⁺ DC1 24h after the last DC injection. (E) Lung index (lung weight/mice weight), cancer score (indicative of the number and size of tumors) and lung total cell number. (F) Lung ratios of CD8 T cell and NK cell numbers relative to the number and size of tumors (cancer score). CD8 T cells were identified as CD45⁺, CD19⁻, CD90.2⁺, CD4⁻, CD8⁺ and NK cells were identified as CD19⁻, CD3ε⁻, B220⁻, CD49b⁺, NK1.1⁺ by flow cytometry analysis. (A) Data are presented as individual dots with means. (B to F) Data are expressed as mean ± SEM. (D, E panel 3) n = 5 representative of two independent experiments. * = p < 0.05, compared as indicated in the graph; φ = p < 0.05 compared to naïve mice. P-values were determined using (B) paired t-test, (D) unpaired t-test and (E-F) one-way ANOVA followed by Tukey's multiple comparisons test.

condition was therefore used to produce the maximal level of cancer-primed CD103$^+$ DC1s in cultures. The phagocytosis of cancer cells by DCs was confirmed in a phagocytosis assay, where B16F10 cells were stained with CFSE prior to the co-stimulation. The CFSE signal was then detected in FLT3L-BMDCs as an indication of phagocytosis. The CFSE MFI signal was increased in DCs exposed to B16F10-CFSE, and was higher in CD103$^+$XCR1$^+$ DCs compared to CD11b$^+$Sirpα$^+$ DCs (Fig 7B and S2B Fig). Therefore we concluded that the stimulation of FLT3L-BMDCs with B16F10 prior injections allows DC1 to be primed with tumor antigens, and used these cells in the injections described in the next section.

Then, DCs were injected alone or in combination with a commonly-used anti-PD-1 [3] in the B16F10 model (Fig 7C), which is reportedly resistant to anti-PD-1 therapies [20]. The injection of XCR1$^+$ DC1s led to a significant increase in the total number of lung CD103$^+$XCR1$^+$ DCs 24h later (Fig 7D). The lung index and cancer score, indicators of the quantity and size of tumors, were significantly increased in all groups injected with B16F10 compared to naïve mice (Fig 7E). The total number of lung cells was significantly increased compared to naïve mice in all groups except for the mice treated with anti-PD-1 in combination with DC injection (Fig 7E). As reported, the treatment with the anti-PD-1 did not impact cancer severity when administered alone. However, combining the anti-PD-1 treatment with DC1 injections significantly decreased the lung index and cancer score compared to the B16F10 control group, and showed a strong tendency to decrease total lung cells, restoring sensitivity to the ICI treatment (Fig 7E). As an indicator of an overall impact of DC transfers on the anticancer response, the number of CD8 T cells and NK cells relative to the number and size of tumors was calculated by dividing the total number of CD8 T cells or NK cells (S2C Fig) by the cancer score, and is presented in Fig 7F as CD8 T cell and NK cell ratios. While the injection of the DC1 alone induced trends towards higher ratios of CD8 T cells and NK cells, the combination of treatments significantly increased the CD8 T cell and NK cell ratios compared to baseline. This suggests that restoring potent anticancer DC populations can not only restore the capacity of anticancer DCs to locally interact with CD8 T cells in response to anti-PD-1s, but also supports the accumulation of CD8 T cells in this context, conferring an even higher potential to this approach.

## Discussion

The clinical efficacy of ICIs is currently undermined by the development of treatment resistance in the long run, in addition to limited patient responsiveness [3, 6]. The presence and activity of pre-existing CD8 T cells that are specific to tumor antigens is an essential condition for the success of ICIs. DCs, particularly DC1s, play a crucial role in priming antigen-specific CD8 T cells [7]. However, the influence of cancer on the local DC lung signature and DC function remains poorly studied. In this report, we precisely report the influence of tumor development on DC populations, with alterations that are likely detrimental to the anticancer immune response.

Our observation that lung tumor development leads to decreased proportions of the anticancer CD103$^+$ DC1 population is supported by various other studies. Indeed, a predominance of the CD11b$^+$ DC2 subset was reported in an orthotopic lung tumor model [33]. Studies using human samples from breast, pancreatic and lung tumors also described decreased proportions of DC1s [34, 35]. Additionally, a decreased DC1 frequency in bone marrow and blood was previously observed in cancer patients [34, 36]. Also, proportions of blood DC1s and the DC1/DC2 ratio were reportedly decreased with cancer severity in lung cancer patients [36]. While supporting our observations, these data also support a high potential to translate this study to human cancer.

Additionally, the association between decreased DC1 proportions and cancer severity suggests that reduced DC1 proportions could be detrimental to the successful activation of anticancer immune responses. While the impairment of the anticancer immune response caused by the genetic abrogation of DC1s was previously described (such as in *Batf3*$^{-/-}$ mouse models) [12–14], the impact of the naturally-occurring decrease in DC1 proportions provoked by lung tumor development (reported here) on the anticancer immune response and cancer severity remained unaddressed. Furthermore, we observed that tumor development skewed the DC response towards DC1s that do not express CD103, which could be due to a decreased CD103 expression on XCR1$^+$ cells; this phenomenon was previously described by us in response to LPS-induced lung inflammation [8]. In the particular case of live cancer cell interaction, we observed that direct contact with DCs was necessary to influence CD103 expression, suggesting that surface molecules expressed by tumor cells are involved. For instance, the interaction between PD-L1 expressed by tumor cells and PD-1 on DCs or the binding of galectin-9 present on tumor cells with TIM-3 expressed on some DC populations, which are known to influence DC functions, could be involved [32, 37]. Further analysis will be necessary to identify the exact mechanism. Additionally, this result suggests that using CD103 as a marker of DC1s could lead to an underestimation of DC1 numbers. Also, and in accordance with our previous publications, this demonstrates that CD103 expression is highly modulable in the lung and should not be used to identify DC1s. Furthermore, the higher expression of regulatory markers PD-L1 and PD-L2 by CD103$^-$XCR1$^+$ DCs likely supports the immunoregulatory tumor microenvironment. Finally, these observations also complement recent reports demonstrating the accumulation of regulatory DC1s in lung cancer [38].

Our analysis of DC populations in lung tumor models also highlighted a CD103$^{lo}$CD11b$^+$ DC population that is not usually observed at these proportions in the lung. Interestingly, most intestine and mesenteric lymph nodes CD11b$^+$ DC2 typically express CD103 under homeostatic and inflammatory conditions [26, 27, 39]. Of note, our results suggest that this population stems from a CD103-expressing DC2 population, similar to CD103$^+$CD11b$^+$ DC populations found in the gut and mesenteric lymph nodes [26, 27]. The only identification marker analyzed in this study that was differentially expressed by CD103$^{lo}$CD11b$^+$ DCs and regular lung CD11b$^+$CD103$^-$ DC2s was the transcription factor IRF4, which was expressed at a higher level in CD103$^{lo}$CD11b$^+$ DCs. This may be due to differential expression with regards to the maturation status of these two populations. A study by Schlitzer *et al.* also observed that in the small intestinal lamina propria, the CD103$^+$CD11b$^+$ DC population expressed higher levels of IRF4 than the CD103$^-$CD11b$^+$ DC populations [40], which strengthens the idea of similarity between lung and gut CD103$^+$CD11b$^+$ DC populations. Several studies demonstrated that intestinal CD103$^+$CD11b$^+$ DCs are crucial in T$_H$17 responses [26, 27]. The development of different cancer types, including NSCLC, is associated with an important increase of T$_H$17 cells in tumors and peripheral blood, but there remains a current lack of consensus in terms of the role of T$_H$17 cells in cancer, as they were alternatively deemed to exert antitumor activity or promote tumor development [41, 42].

While few studies reported a CD103$^{+/lo}$CD11b$^+$ DC population in the lung, Sharma et *al.* observed a CD103$^+$Ly-6C$^+$ DC population that expressed CD11b in tumor extracts from mice treated with chemotherapy. However, this DC population was different from the CD103$^{lo}$CD11b$^+$ observed here, as it expressed DC1-associated markers and no PD-L1. [43]. Another study also reported an accumulation of a CD103$^+$CD11b$^+$ DC population in the lung, in a mouse model of infection with *H. capsulatum* treated with an anti-TNF. While they did not fully characterize this DC population, authors report its involvement in regulatory T cell amplification, similar to what was previously reported for CD103$^+$CD11b$^+$ gut DCs [44, 45]. An interesting study published by Maier *et al.* using single-cell RNA sequencing in a murine

model of lung adenocarcinoma lesions recently identified a DC population with high levels of maturation markers such as CD80, CD86 and MHC II, and also immunoregulatory genes *Pdcd1lg2* (PD-L2 gene) and *Cd200*. They consequently named this population 'mature DCs enriched in immunoregulatory molecules' (mregDCs). In addition, a large proportion of their mregDC population expresses both CD103 and CD11b. While in our hands the CD103$^{lo}$CD11b$^+$ DC population is mainly associated with DC2s, their mregDCs include cells from both DC1 and DC2 populations, which suggest that the population we identified might be a subpopulation of total mregDCs as identified in this study [38].

Lung DCs originate from either bone marrow-derived pre-DCs or monocytes [11, 28]. Our analysis of CCR2 and Ly-6C expression, both associated with the monocyte-derived DC lineage [28, 29], and our analysis of ZBTB46, a marker of conventional DCs and pre-DCs [30] suggests that this population originates from pre-DCs. Of note, intestinal CD103$^+$CD11b$^+$ DCs also originate from pre-DCs [39]. Several mechanisms could explain the accumulation of lung CD103$^{lo}$CD11b$^+$ DC following tumor development. Pre-DCs could be directly recruited from bone marrow to the lung and then differentiated into CD103$^{lo}$CD11b$^+$ DCs. As GM-CSF induces CD103, high levels of GM-CSF in the lung in cancer could also induce local CD103 expression on CD11b$^+$ DC2s [8, 23, 46]. As the fate of DCs (DC1 vs DC2 lineage) is determined at the progenitor stage [47], it would be interesting to determine whether this population is associated with a specific bone marrow progenitor. Another hypothesis to explain the presence of CD103$^{lo}$CD11b$^+$ DCs is that lung tumor development could induce the migration of CD103$^+$CD11b$^+$ DCs from the gut to the lung, as was suggested in an *H. capsulatum* infection model [44].

Our analysis of several surface proteins involved in DC functions revealed that lung CD103$^{lo}$CD11b$^+$ DCs could potentially influence T cell responses and ultimately the efficiency of anticancer immune response. Indeed, the interaction between PD-1 and its ligands PD-L1 and PD-L2 leads to the inhibition of T cells [5]. Furthermore, the blockade or silencing of PD-L1 or PD-L2 on DCs results in higher production of IL-12 and enhanced DC maturation combined with improved T cell antitumor function [48, 49]. Also, PD-L1 expression on antigen-presenting cells correlated with clinical efficacy of PD-L1 and PD-1 blockade in a cohort of melanoma patients [50]. These results suggest that the observed high expression of both PD-L1 and PD-L2 on CD103$^{lo}$CD11b$^+$ DCs could restrain T cell anticancer responses. This is of major importance to the field, as we have identified a DC population that could counteract the positive impact of DC1s and that may explain the failure of DC1s to naturally control tumor development in the lung in cancer. A better understanding of the origin of this population could lead to strategies controlling their recruitment to ultimately modulate anticancer immune responses.

Several studies report the presence of regulatory DCs in tumor environment [38, 51–53]. Yet, there is actually no consensus on what defines "regulatory DC" in cancer. Some suggest that immature DCs possess regulatory or tolerogenic functions, since they inhibit innate and adaptive immune responses [51, 52]. Others claimed that "regulatory DCs" produce high levels of anti-inflammatory cytokines, and are involved in regulatory T cell development [54]. Finally, high expression of regulatory molecules like PD-1 or PD-L1 is also attributed to "regulatory DCs" [38, 51, 53]. It is very likely that currently, cells under the "regulatory DCs" label comprise different subtypes of DCs at different developmental or maturation stages. Importantly, we and others demonstrate that the expression of some surface proteins like CD103 and PD-L1 can be modulated in various contexts [8, 53]. Therefore, an approach based on the analysis of transcription factors involved in DC development like BATF3, IRF8 and IRF4, combined to surface markers stably expressed from the progenitor to mature stage of DCs would help better characterize regulatory DC populations. In any case, we feel the strong expression

of regulatory molecules on CD103$^{lo}$CD11b$^{+}$ DCs reported in our cancer models allows the classification of this DC population under the "regulatory DC" scope, as one of the best-characterized regulatory DC populations in cancer to date, and a possible new explanation of the failure of antitumor DC1s to control cancer spread.

It is widely recognized that CD103$^{+}$ DC1s are important to the anticancer T cell response [13, 14]. In this report, we observed that lung tumors development leads to altered proportions of CD103$^{+}$ DC1s populations. This may, in combination with the accumulation of CD103$^{lo}$CD11b$^{+}$ regulatory DCs, explain the mitigated impact of current immunotherapies. Indeed, even if ICIs therapies re-establish the capacity of T cells to induce effective antitumor responses, it remains that the local DC population is ill-equipped to present cancer antigen, which could be circumvented by injection of *ex-vivo* conditioned autologous DCs in humans. Interestingly, the vast majority of previous clinical trials using DCs as a cancer vaccination approach used mo-DCs, which are functionally different from CD103$^{+}$ DC1s [55]. Our study strongly argues that the injection of purified DC1s could improve the success of DC vaccination therapies in cancer. In both mice and humans, *in vitro* differentiation of DCs in a media containing FLT3L generates a large number of DC1s [23, 56]. To date, few studies have tested the anticancer efficiency of FLT3L-derived DCs. Interestingly, FLT3L injections, which strongly increase the number of circulating DC, improve ICIs efficacy in different mouse models of cancer supporting an important anticancer potential for FLT3L-DCs. [17, 57]. Furthermore, in models of subcutaneous cancer, the injection of FLT3L-BMDCs reduced the progression of tumor volume [21, 58]. These results propose an interesting therapeutic potential for FLT3L-BMDCs. However, in our hands, the DC-alone treatment in the B16F10 model of lung metastasis did not prove effective, which likely relates to the aggressiveness of the model and the consequent strong *in vivo* immunosuppressive environment altering the anticancer potential of local DC populations. Therefore, tackling the presence of immunosuppressive molecules and cells (such as CD103$^{lo}$CD11b$^{+}$ DC2s reported here) should remain a priority to fully address the potential of DC transfers to treat cancer.

One technique commonly used to improve the efficacy of DC injections is to stimulate DCs with tumor antigens [20, 55]. Here, despite an important production of IL-12 following the stimulation of DCs with live B16F10 cells, the injection of DCs alone did not reduce cancer severity. Several studies have demonstrated that DC stimulation with TLR agonists improves their anticancer efficiency [20, 21]. It therefore might be interesting to use this method to improve anticancer function of FLT3L-BMDCs.

New trends in cancer treatment combine two or more immunotherapies [3]. Based on this approach, and considering the fact that the B16F10 melanoma model is resistant to anti-PD-1 alone [20] and that we observed a deficiency in potent anticancer DC1s in this model, we combined FLT3L-BMDC injections to the anti-PD-1, to restore, at least partially, a potent DC1 population in the lung at the time of PD-1/PD-L1 axis blockade. Our results suggest that the optimal ICIs response required the presence of efficient anticancer DC populations. This is of high interest to the field, as DC populations could be evaluated in patients and restored whenever needed to improve the response to ICI therapies.

In conclusion, using mouse models of lung cancer and lung melanoma metastasis, we demonstrate that lung tumor development significantly modulates DC populations at the expense of antitumor DC1s, favouring an unusual accumulation of CD103$^{lo}$CD11b$^{+}$ DC2s that express regulatory molecules. We also demonstrate that enriching the local DC population with CD103$^{+}$ DC1s restores the efficacy of anti-PD-1 therapy. These results suggest that, despite mitigated previous clinical trials using DC vaccination, targeting DC populations remains a valid therapeutic approach to favour the anticancer immune response or to improve existent ICI therapies in lung cancer.

## Supporting information

**S1 Fig. Gating strategy for the identification of lung DC populations.** Gating strategy for the identification of DCs, associated with Fig 4. Total DCs were gated on auto-fluorescence[-], NK1.1[-], CD90.2[-], CD19[-], MHC II[Hi] and CD11c[+]. For this Fig, three populations were segregated prior to the further analysis of DC1 and DC2 markers, i,e CD103[+]CD11b[-/lo] (green), CD11b[+]CD103[-] (blue) and CD103[lo]CD11b[+] DCs (red). A representative flow cytometry histogram showing the normalized number of cells (unit area) on the Y-axis and fluorescence intensity on the X-axis is presented for the XCR1, Sirpα, IRF8, IRF4, CCR2, Ly-6C and ZBTB46 expression for each of these three DC populations. FMO controls appear in grey in each histogram.
(TIF)

**S2 Fig. *In vitro* and *in vivo* characterization of FLT3L-BMDCs generated for transfer experiments.** (A) FLT3L-BMDCs were stimulated with GM-CSF (Day 0). Either live B16F10/ LLC cells or a B16F10/ LLC antigenic preparation was added on day 0 or day 2 and the percentage of CD103[+]XCR1[+] DCs was measured by flow cytometry on day 3. Data are presented as individual dots with means. n = 4 pooled from two independent experiments. (B) Two days following GM-CSF stimulation FLT3L-BMDCs were stimulated for 24h with CFSE-treated B16F10 cells and Δ CFSE MFI (CFSE MFI of DC exposed to CFSE-B16F10 –CFSE MFI of unexposed DC (negative control)) in CD103[+]XCR1[+] DC1 and CD11b[+]Sirpα[+]DC2 and the percentage of CD103[+]XCR1[+] DC1 and CD11b[+]Sirpα[+]DC2 of the total CFSE[+] population was measured by flow cytometry. Data are expressed as mean ± SEM. n = 6, pooled from two independent experiments. [*] = $p < 0.05$ using a paired t-test. (C) The total number of lung CD8 T cells and NK cells 18 days following B16F10 injections. CD8 T cells were identified as CD45[+], CD19[-], CD90.2[+], CD4[-], CD8[+] and NK cells were identified as CD19[-], CD3e[-], B220[-], CD49b[+], NK1.1[+] by flow cytometry analysis. Data are expressed as mean ± SEM. n = 9–11 pooled from two independent experiments. ϕ = $p < 0.05$ compared to naïve mice. P-values were determined using a one-way ANOVA followed by Tukey's multiple comparisons test.
(TIF)

## Acknowledgments

The authors would like to thank the CRIUCPQ animal unit as well as Marc Veillette, head of the flow cytometry platform, as well as the CRIUCPQ animal unit personnel for their precious collaboration and support.

## Author Contributions

**Conceptualization:** Julyanne Brassard, Philippe Joubert, David Marsolais, Marie-Renée Blanchet.

**Formal analysis:** Julyanne Brassard.

**Funding acquisition:** Marie-Renée Blanchet.

**Investigation:** Julyanne Brassard, Meredith Elizabeth Gill, Emilie Bernatchez, Véronique Desjardins, Joanny Roy.

**Methodology:** Julyanne Brassard, Emilie Bernatchez, Marie-Renée Blanchet.

**Supervision:** Marie-Renée Blanchet.

**Writing – original draft:** Julyanne Brassard.

**Writing – review & editing:** Julyanne Brassard, Emilie Bernatchez, Philippe Joubert, David Marsolais, Marie-Renée Blanchet.

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
