## [Decision Letter · Decision Letter 0]

16 Aug 2021

Bélatelep, Hungary

August 11, 2021

PONE-D-21-20655

Countering the advert effects of lung cancer on the anticancer potential of dendritic cell populations reinstates sensitivity to anti-PD-1 therapy.

PLOS ONE

Dear Dr. Blanchet,

Thank you for submitting your manuscript to PLOS ONE. After careful consideration, we feel that it has merit but does not fully meet PLOS ONE’s publication criteria as it currently stands. Therefore, we invite you to submit a revised version of the manuscript that addresses the points raised by  the Reviewers, listed below.

We look forward to receiving your revised manuscript.

Kind regards,

Joseph Najbauer, Ph.D.

Academic Editor

PLOS ONE

Journal Requirements:

3. We note that Figure (7) in your submission contain copyrighted images. All PLOS content is published under the Creative Commons Attribution License (CC BY 4.0), which means that the manuscript, images, and Supporting Information files will be freely available online, and any third party is permitted to access, download, copy, distribute, and use these materials in any way, even commercially, with proper attribution. For more information, see our copyright guidelines: http://journals.plos.org/plosone/s/licenses-and-copyright.

1. You may seek permission from the original copyright holder of Figure (7) to publish the content specifically under the CC BY 4.0 license. 

Reviewers' comments:

Reviewer's Responses to Questions

**Comments to the Author**

1. Is the manuscript technically sound, and do the data support the conclusions?

Reviewer #1: Yes

Reviewer #2: Yes

2. Has the statistical analysis been performed appropriately and rigorously? 

Reviewer #1: Yes

Reviewer #2: Yes

3. Have the authors made all data underlying the findings in their manuscript fully available?

Reviewer #1: Yes

Reviewer #2: Yes

4. Is the manuscript presented in an intelligible fashion and written in standard English?

Reviewer #1: Yes

Reviewer #2: No

5. Review Comments to the Author

Reviewer #1: In the manuscript “Countering the advert effects of lung cancer on the anticancer potential of dendritic cell populations reinstates sensitivity to anti-PD-1 therapy.”, two mice models have been used to show the impact of cancer on dendritic cells (DC) in the lung. The authors conclude that cancer can change the composition of DC, with the generation of a CD103low DC population with DC2 markers and high levels of T-cell inhibitory molecules. Injection of DC1 in combination with anti-PD-1 checkpoint inhibitor could restore sensitivity to immunotherapy. The manuscript is well organized and written. The authors should consider the following comments:

1. The authors describe the relative proportions of DC subpopulations. The proportion of CD103+ DC with anticancer functions is falling whereas the proportion of CD11b+ DC with a low CD103 expression is rising. However, what does this mean for absolute DC numbers in the lung? Could the authors use microscopy or another method to get a score of total DC/lung?

2. What happens in the blood of the tumor-exposed mice? The authors should show whether one can find CD103low DC in the blood. This would have the advantage that one can give absolute numbers of DC subpopulations over time. Could one use the time course of CD103low DC as a biomarker?

3. The authors have to explain their flow cytometry experiments in more detail. Which channel was used for “autofluorescence”? Which viability marker was used? How many cells were measured to get enough DC? Did you identify and exclude doublets?

4. If direct contact between tumor cells and DC changes the DC subpopulations of the lung, which tumor molecules could be involved in the process?

Reviewer #2: Below are some questions and remarks to the authors

General

The manuscript should be carefully proofread and adapted with special attention to English grammar. Several sentences require reformulation to bring the message more clear and scientifically correct. The choice of words and the lack of punctuations makes reading more difficult.

Introduction

In the introduction the authors write about lung cancer, the different DC subsets and their findings. Although it is a coherent introduction, the authors should elaborate more on immune checkpoints and their relevance in lung cancer especially, the PD-1/PD-L1 and PD-L2 axis.

Material and Methods

The phagocytosis assay (line 132) should be described under a separate subtitle as it is a read-out (proof of phagocytosis by DC) and not a production method of the FLT3L-BMDCs.

Results

The DC1s, CD103LoCD11b+ DCs and DC2s are phenotypically characterized by the authors. Based on the phenotype, various assumptions about their functionality are made. These assumptions could be confirmed by performing functional assays comparing the different subsets of the DCs. Do the authors have functional data or could they perform some functional assays, such as T cell activation and proliferation assays or a migration assay?

In 4.2 the authors refer to a figure with “(Figure 2A and B)” in other cases the authors refer with “(Figure 2DH )”. Use consistency in your references.

In the caption of figure 2 line 254-256, the authors write that figure 2F, G, H and I represent the percentage and ΔMFI of PD-L1, PD-L2, CD200 and MHC II of DC subtypes in naïve mice and mice injected with B16F10 cells. The graphs F, G, H and I only show data of DC subtypes in mice injected with B16F10 cells. Please add the data of naïve mice or correct the caption.

6. PLOS authors have the option to publish the peer review history of their article (what does this mean?). If published, this will include your full peer review and any attached files.

Reviewer #1: No

Reviewer #2: No

---

## [Author Response · Author response to Decision Letter 0]

6 Oct 2021

Reviewer 1 comments: 

Reviewer #1: In the manuscript “Countering the advert effects of lung cancer on the anticancer potential of dendritic cell populations reinstates sensitivity to anti-PD-1 therapy.”, two mice models have been used to show the impact of cancer on dendritic cells (DC) in the lung. The authors conclude that cancer can change the composition of DC, with the generation of a CD103low DC population with DC2 markers and high levels of T-cell inhibitory molecules. Injection of DC1 in combination with anti-PD-1 checkpoint inhibitor could restore sensitivity to immunotherapy. The manuscript is well organized and written. The authors should consider the following comments:

1. The authors describe the relative proportions of DC subpopulations. The proportion of CD103+ DC with anticancer functions is falling whereas the proportion of CD11b+ DC with a low CD103 expression is rising. However, what does this mean for absolute DC numbers in the lung? Could the authors use microscopy or another method to get a score of total DC/lung?

2. What happens in the blood of the tumor-exposed mice? The authors should show whether one can find CD103low DC in the blood. This would have the advantage that one can give absolute numbers of DC subpopulations over time. Could one use the time course of CD103low DC as a biomarker?

3. The authors have to explain their flow cytometry experiments in more detail. Which channel was used for “autofluorescence”? Which viability marker was used? How many cells were measured to get enough DC? Did you identify and exclude doublets?

4. If direct contact between tumor cells and DC changes the DC subpopulations of the lung, which tumor molecules could be involved in the process?

Response to reviewer 1 comments:

1. We thank R1 for this pertinent question. All DC populations are increased in cancer, but the proportions of each population are, as reported, very different. Therefore, even if DC1s increase in numbers following cancer development, they are dramatically overtaken by DC2s in this context, which also increase in numbers, but un much higher proportions. Also, anticancer DC1s are found in a 1:1 ratio with regulatory CD103lo/CD11b+ DCs following tumour development. Taking this into consideration, it becomes clear that the data which speaks the most as to the nature of the DC response in the lung remains DC proportions. We hope this clarifies our reasoning for not including total numbers in figures. 

2. This is an interesting question. Quickly, precursors do not express CD103 in the blood, so CD103 low DCs as a biomarker could be used, but would need to come from the lung. 

3. We thank R1 for this important question, and have now provided the information in the text. 

4. We thank the reviewer for this interesting suggestion and have provided additional information with regards to potential tumour molecules interactions in the text. 

Reviewer 2 comments:

General

The manuscript should be carefully proofread and adapted with special attention to English grammar. Several sentences require reformulation to bring the message more clear and scientifically correct. The choice of words and the lack of punctuations makes reading more difficult.

Introduction

In the introduction the authors write about lung cancer, the different DC subsets and their findings. Although it is a coherent introduction, the authors should elaborate more on immune checkpoints and their relevance in lung cancer especially, the PD-1/PD-L1 and PD-L2 axis.

Material and Methods

The phagocytosis assay (line 132) should be described under a separate subtitle as it is a read-out (proof of phagocytosis by DC) and not a production method of the FLT3L-BMDCs.

Results

The DC1s, CD103LoCD11b+ DCs and DC2s are phenotypically characterized by the authors. Based on the phenotype, various assumptions about their functionality are made. These assumptions could be confirmed by performing functional assays comparing the different subsets of the DCs. Do the authors have functional data or could they perform some functional assays, such as T cell activation and proliferation assays or a migration assay?

In 4.2 the authors refer to a figure with “(Figure 2A and B)” in other cases the authors refer with “(Figure 2DH )”. Use consistency in your references.

In the caption of figure 2 line 254-256, the authors write that figure 2F, G, H and I represent the percentage and ΔMFI of PD-L1, PD-L2, CD200 and MHC II of DC subtypes in naïve mice and mice injected with B16F10 cells. The graphs F, G, H and I only show data of DC subtypes in mice injected with B16F10 cells. Please add the data of naïve mice or correct the caption.

Response to reviewer 2 comments:

General: we appreciate R2’s feedback on the quality of grammar in the text, and have consequently carefully reviewed the text. 

Introduction: We have added information of PD-L1 therapy as proposed by the reviewer. 

Methods: we thank R2 for this useful suggestion and have provided the information in a separate paragraph as suggested. 

Results: 

- We agree with R2 that performing functional assays on CD103lo CD11b DCs would be interesting. We have considered this often, and are unfortunately limited by the low total number of DCs which can be isolated from lung. We have evaluated that we would need over 10 mice per condition to have enough cells to perform functional assays, which, ethically, is not acceptable for our local committee. We therefore try to circumvent this technical limitation by adjusting our message and insuring that we discuss the regulatory “potential” of this population rather than affirming they possess a regulatory function. We hope this explanation will satisfy the reviewer but understand their feedback. Of note, many publications seem limited by this technical limitation with regards to regulatory DCs, and publish regulatory molecule expression as a marker of DC regulatory potential. 

- We thank R2 for the feedback on consistency in figure naming and have adjusted the text accordingly. 

- The caption was corrected as suggested by R2.

---

## [Decision Letter · Decision Letter 1]

15 Nov 2021

Countering the advert effects of lung cancer on the anticancer potential of dendritic cell populations reinstates sensitivity to anti-PD-1 therapy.

PONE-D-21-20655R1

Dear Dr. Blanchet,

We’re pleased to inform you that your manuscript has been judged scientifically suitable for publication and will be formally accepted for publication once it meets all outstanding technical requirements.

Please correct a very minor typo, in Materials and. Methods, you indicate that  you are using "Tukey's multiple comparison tests" which is correct. However, in the mansucript you write Turkey's multiple comparisons tests in many figure legends.

Kind regards,

Jean Kanellopoulos, M.D., Ph.D.

Academic Editor

PLOS ONE

Additional Editor Comments (optional):

Reviewers' comments:

Reviewer's Responses to Questions

**Comments to the Author**

1. If the authors have adequately addressed your comments raised in a previous round of review and you feel that this manuscript is now acceptable for publication, you may indicate that here to bypass the “Comments to the Author” section, enter your conflict of interest statement in the “Confidential to Editor” section, and submit your "Accept" recommendation.

Reviewer #2: All comments have been addressed

2. Is the manuscript technically sound, and do the data support the conclusions?

Reviewer #2: Yes

3. Has the statistical analysis been performed appropriately and rigorously? 

Reviewer #2: Yes

4. Have the authors made all data underlying the findings in their manuscript fully available?

Reviewer #2: Yes

5. Is the manuscript presented in an intelligible fashion and written in standard English?

Reviewer #2: Yes

6. Review Comments to the Author

Reviewer #2: (No Response)

7. PLOS authors have the option to publish the peer review history of their article (what does this mean?). If published, this will include your full peer review and any attached files.

Reviewer #2: No

---

## [Editor Report · Acceptance letter]

18 Nov 2021

PONE-D-21-20655R1 

Countering the advert effects of lung cancer on the anticancer potential of dendritic cell populations reinstates sensitivity to anti-PD-1 therapy. 

Dear Dr. Blanchet:

I'm pleased to inform you that your manuscript has been deemed suitable for publication in PLOS ONE. Congratulations! Your manuscript is now with our production department. 

Kind regards, 

on behalf of

Dr. Jean Kanellopoulos 

Academic Editor

PLOS ONE